# Efficiency of Interferon-γ in Activating Dendritic Cells and Its Potential Synergy with Toll-like Receptor Agonists

**DOI:** 10.3390/v15051198

**Published:** 2023-05-19

**Authors:** Yuanzhi Bian, Debra L. Walter, Chenming Zhang

**Affiliations:** Department of Biological Systems Engineering, College of Agriculture and Life Sciences & College of Engineering, Virginia Tech, Blacksburg, VA 24061, USA

**Keywords:** interferon-γ, dendritic cell activation, synergy, TLR agonist, immunotherapy, vaccine, antigen presentation, flow cytometry

## Abstract

Interferon-γ (IFN-γ) is a cytokine that plays an important role in immune regulation, especially in the activation and differentiation of immune cells. Toll-like receptors (TLRs) are a family of pattern-recognition receptors that sense structural motifs related to pathogens and alert immune cells to the invasion. Both IFN-γ and TLR agonists have been used as immunoadjuvants to augment the efficacy of cancer immunotherapies and vaccines against infectious diseases or psychoactive compounds. In this study, we aimed to explore the potential of IFN-γ and TLR agonists being applied simultaneously to boost dendritic cell activation and the subsequent antigen presentation. In brief, murine dendritic cells were treated with IFN-γ and/or the TLR agonists, polyinosinic–polycytidylic acid (poly I:C), or resiquimod (R848). Next, the dendritic cells were stained for an activation marker, a cluster of differentiation 86 (CD86), and the percentage of CD86-positive cells was measured by flow cytometry. From the cytometric analysis, IFN-γ efficiently stimulated a considerable number of the dendritic cells, while the TLR agonists by themselves could merely activate a few compared to the control. The combination of IFN-γ with poly I:C or R848 triggered a higher amount of dendritic cell activation than IFN-γ alone. For instance, 10 ng/mL IFN-γ with 100 µg/mL poly I:C achieved 59.1% cell activation, which was significantly higher than the 33.4% CD86-positive cells obtained by 10 ng/mL IFN-γ. These results suggested that IFN-γ and TLR agonists could be applied as complementary systems to promote dendritic cell activation and antigen presentation. There might be a synergy between the two classes of molecules, but further investigation is warranted to ascertain the interaction of their promotive activities.

## 1. Introduction

Adjuvants are substances applied in vaccine formulations or cancer immunotherapies to enhance immune responses. They activate antigen-presenting cells (APCs) and facilitate antigen presentation through the upregulated expression of major histocompatibility complexes (MHCs) [1,2]. APCs, including dendritic cells, macrophages, and B cells, are considered the bridge between innate and adaptive immunity. They internalize pathogens or tumor cells through phagocytosis or receptor-mediated endocytosis. The pathogens or tumor cells are then degraded in endosomal–lysosomal compartments within APCs by proteases. The pathogenic or tumorigenic antigens are taken up by MHC molecules, which ultimately migrate to the surface of APCs to interact with T cell receptors on CD4+ or CD8+ T cells [3,4]. Over the past years, subunit vaccines have gained popularity due to high antigen purity and improved safety compared to whole-organism vaccines. However, subunit vaccines are often limited by low immunogenicity because they lack intrinsic elements to stimulate APCs. In this case, adjuvants are complemented in the vaccine formulations to activate immune responses more efficiently [5,6]. In another case, vaccines against small molecules that are not or weakly immunogenic, such as nicotine and oxycodone, require delivery vehicles and adjuvants to obtain the desired immunogenicity [7,8,9]. There are various types of adjuvants, such as aluminum salt-based adjuvants, emulsion adjuvants, and APC surface receptor agonists [10]. In this study, we focused on studying the efficiency of cytokines and toll-like receptor (TLR) agonists in activating dendritic cells, which are the dominant class of APCs. Upon activation, dendritic cells start to express MHC molecules and migrate to lymph nodes, where they interact with T cells [11,12]. As both MHC expression and DC migration are prerequisites for antigen presentation to take place, the increased activation of dendritic cells may translate to more efficient antigen presentation.

Cytokines are small secreted proteins involved in a wide range of cell communication and signaling pathways, especially in the stimulation and regulation of immune responses [13]. Multiple classes of cytokines have been identified, including the interleukin (IL) family, tumor necrosis factors, and interferons. Among these, interferon-γ (IFN-γ) holds great promise for being used as an immunoadjuvant to boost the efficacy of immunotherapies [14,15]. IFN-γ is primarily produced by activated T cells and natural killer (NK) cells in response to certain antigens or other cytokines. It plays an important role in immune regulation, especially in the activation and differentiation of immune cells, including T cells, B cells, NK cells, and macrophages [16]. In addition, IFN-γ upregulates the expression of MHC class II molecules, therefore promoting antigen presentation to CD4+ T cells [4,17]. Because of these immunostimulatory effects, IFN-γ has been investigated in vaccines against various human and animal infectious diseases. For instance, IFN-γ was shown to increase the humoral response against hemagglutinin and neuraminidase surface proteins in mice immunized with influenza subunit vaccines [18]. Expression of IFN-γ in conjunction with HIV-1 gp120 produced stronger primary antibody and T-cell responses to the gp120 protein than the vaccination with gp120 alone [19]. Immunization of ducks with an IFN-γ expressed vector and a duck hepatitis B virus (DHBV) vaccine has been shown to increase the protection of animals against DHBV infection [20].

TLR agonists have been considered a potent class of adjuvants that improves the efficacy of cancer immunotherapies [21,22] and vaccines against infectious diseases [23,24]. TLRs are a family of pattern recognition receptors (PRRs) that serve as primary sensors of innate immunity. They recognize distinct structural motifs related to pathogens or components of host cells released during cell damage, often referred to as pathogen-associated molecular patterns (PAMPs) or damage-associated molecular patterns (DAMPs), respectively. TLRs are expressed on immune cells (including dendritic cells, macrophages, granulocytes, T cells, B cells, NK cells, and mast cells), endothelial and epithelial cells, as well as tumor cells. Some TLRs are present on the plasma membrane (TLR1, 2, 4, 5, and 6), while others are located within the endoplasmic reticulum and rapidly recruited to endosomal–lysosomal compartments upon pathogen invasion or host cell death (TLR3, 7, 8, and 9) [21,22]. Ligand binding to TLRs triggers a cascade of signaling pathways that enhances the secretion of cytokines, stimulates the maturation of APCs, and boosts the production of antigen-specific antibodies. TLR agonist-incorporated cancer immunotherapies and vaccines have demonstrated their capability of eliciting a stronger immune response than TLR agonist-free counterparts [9,21,22].

Upon TLR ligand binding, either the toll/IL-1R domain-containing adaptor-inducing IFN-β (TRIF)- or the myeloid differentiation factor 88 (MyD88)-dependent signaling pathway is activated. These pathways lead to the activation of the transcription factors nuclear factor-kappa B (NF-κB) or interferon-regulatory factors (IRFs) to regulate the expression of cytokines, such as type I interferons [25,26,27]. Type I interferons include IFN-α, IFN-β, IFN-δ, IFN-ε, IFN-κ, IFN-τ, and IFN-ω; IFN-γ, on the other hand, is the only type II interferon. Type I and type II interferons are distinguished by the fact that they bind different receptors. All type I interferons bind the same receptor named the type I interferon receptor, while IFN-γ binds a different one known as the type II interferon receptor [28]. The downstream signal transduction pathways associated with the receptors are slightly different. Type I interferons lead to the activation of the transcription factors designated as interferon-stimulated response elements (ISREs), while IFN-γ activates gamma-activated sequence (GAS) promoter elements. ISREs and GAS promoter elements together supervise the expression of a collection of genes called the interferon-stimulated genes, which are mostly antiviral genes [29,30,31]. Both type I interferons and IFN-γ have been demonstrated to promote dendritic cell maturation [32]; therefore, TLR agonists that result in the production of type I interferons might possess synergistic potential with IFN-γ to augment dendritic cell activation.

While many studies have investigated the efficacy of using either IFN-γ or TLR agonists as immunoadjuvants, the beneficial effects of combining them are underexplored. Additionally, there is currently a dearth of studies that directly compare the efficiency of adjuvants or combinations of adjuvants in activating dendritic cells. Dendritic cells are the most prominent APCs compared to the others and uniquely able to promote naïve T cell activation and effector differentiation [33,34]. Hence, this study aimed to explore and compare the efficiency of the combination of IFN-γ and TLR agonists in inducing dendritic cell activation. In brief, murine dendritic cells were treated with IFN-γ and/or the TLR agonists, polyinosinic–polycytidylic acid (poly I:C; a TLR3 agonist) or resiquimod (R848; a TLR7/8 agonist). The dendritic cells were then stained for a cluster of differentiation 86 (CD86), an activation marker of dendritic cells, using a phycoerythrin (PE) linked anti-CD86 antibody. The percentage of CD86-positive cells, representative of dendritic cell activation level, was analyzed on a flow cytometer. In addition, the toxicity of IFN-γ and the TLR agonists was assessed by measuring the percentage of living cells post-treatment. The percentages of the treatment groups were compared to that of the control, and any significant difference was deemed to be suggestive of potential toxicity. The treatment that demonstrated great efficacy in activating dendritic cells and minimal toxic activity could be further studied as promising adjuvants in cancer immunotherapies or vaccines against infectious diseases or psychoactive compounds.

## 2. Materials and Methods

### 2.1. Materials

JAWSII (ATCC^®^ CRL-11904™) murine dendritic cells and fetal bovine serum (FBS) were purchased from ATCC (Manassas, VA, USA). Because it is an immortalized, commercially available cell line, the use of JAWSII dendritic cells was exempted from approval by Virginia Tech Institutional Animal Care and Use Committee. Gibco™ alpha minimum essential medium (MEM α) containing ribonucleosides, deoxyribonucleosides, and L-glutamine, Gibco™ recombinant murine granulocyte-macrophage colony-stimulating factor (GM-CSF), Gibco™ 0.25% Trypsin-EDTA solution, Invitrogen™ TRIzol™ Reagent, Invitrogen™ MultiScribe™ Reverse Transcriptase, 1 mg/mL 4′,6-diamidino-2-phenylindole (DAPI) solution, and bovine serum albumin (BSA) were purchased from ThermoFisher Scientific (Waltham, MA, USA). Phycoerythrin (PE) linked anti-mouse CD86 antibody and flow staining buffer were purchased from BioLegend (San Diego, CA, USA). Poly I:C sodium salt and gelatin from porcine skin were purchased from Sigma–Aldrich (St. Louis, MO, USA). R848 was purchased from InvivoGen (San Diego, CA, USA). Recombinant mouse IFN-γ was purchased from BD Biosciences (Franklin Lakes, NJ, USA).

### 2.2. Culturing JAWSII Murine Dendritic Cells

JAWSII murine dendritic cells were cultured according to the manufacturer’s instructions. In brief, the cells were seeded into a cell culture flask containing 10 mL of complete growth medium. The complete growth medium was composed of 80% MEM α, 20% FBS, and 5 ng/mL murine GM-CSF. The cells were allowed to grow to ~90% confluency before being passaged or harvested for experimentation. When subculturing the cells, 0.25% Trypsin-EDTA solution was applied to detach the cells from the tissue culture flask. Detached cells were collected by centrifugation at 105× *g* for 10 min. Pooled cells were then resuspended and diluted in 10 mL of fresh complete growth medium.

### 2.3. Verifying the Expression of TLRs

Reverse transcription-polymerase chain reaction (RT-PCR) was performed to verify the expression of TLRs by JAWSII dendritic cells. Total RNA was extracted from the dendritic cells using TRIzol™ Reagent. In short, TRIzol™ Reagent was applied to lyse the cells, and chloroform was added to separate the cell lysate into different layers. The total RNA resided on the clear upper aqueous layer. The aqueous phase was treated with isopropanol and centrifuged to precipitate the RNA. The RNA was washed and reverse transcribed to form complementary DNA (cDNA) molecules using MultiScribe™ Reverse Transcriptase. Subsequently, short sequences (amplicons) on the cDNA characteristic of the genes of TLR3, 4, 7, and 8 were amplified by PCR. The PCR products were then examined by gel electrophoresis.

### 2.4. Determining the Time to Treat Dendritic Cells 

The optimum time to treat the dendritic cells with IFN-γ and the TLR agonists was determined based on the amount of cell activation observed at various time points of stimulation. In brief, the cells were serum-starved in a growth medium containing 0.2% serum for 6 h to synchronize them to the same cell cycle phase [35]. The cells were then treated with 100 µg/mL poly I:C together with 10 ng/mL IFN-γ for 2, 4, 12, 18, or 24 h. Subsequently, the cells were stained for CD86, an activation marker of dendritic cells, using the PE anti-mouse CD86 antibody. DAPI was used as a cell viability stain to monitor any toxic effects of the treatment. Dendritic cell activation, as a result of poly I:C and IFN-γ stimulation, was quantitated by the percentage of CD86-positive cells measured by flow cytometry. The time point or period that resulted in the highest amount of cell activation was selected as the treatment time for the following flow cytometry experiments. 

### 2.5. Evaluating the Efficiency of IFN-γ and TLR Agonists in Activating Dendritic Cells

The activation of the dendritic cells by IFN-γ or the TLR agonists, poly I:C or R848, was studied by flow cytometry. JAWSII dendritic cells (2 × 10^6^) were seeded in a 6-well plate and allowed to adhere overnight. Prior to stimulation, the cells were placed in the growth medium containing 0.2% serum for 6 h. The dendritic cells were then incubated with one of the aforementioned molecules for 12–18 h. The negative control used in this and the following flow cytometry experiments were JAWSII dendritic cells treated with the serum starvation medium, which was 99.8% MEM α and 0.2% FBS. In addition to bringing the dendritic cells to the same cell cycle phase, the serum starvation medium was also the vehicle in which IFN-γ and the TLR agonists were delivered. After treatment, the cells were washed three times with PBS and detached from the wells using a 0.25% Trypsin-EDTA solution. The detached cells were washed twice with flow staining buffer and stained for CD86 using the PE anti-mouse CD86 antibody. The percentage of living cells was investigated by staining the cells with DAPI. Finally, the stained cells were analyzed on a FACSAria™ Fusion Flow Cytometer made by BD Biosciences (San Jose, CA, USA).

### 2.6. Exploring the Potential of IFN-γ and TLR Agonists as Complementary Systems

The activation of the dendritic cells by IFN-γ together with one of the TLR agonists, poly I:C or R848, was examined by flow cytometry. JAWSII dendritic cells (2 × 10^6^) were seeded in a 6-well plate and allowed to adhere overnight. The cells were first placed in the serum starvation medium for 6 h to be synchronized to the same cell cycle phase. The dendritic cells were then treated with IFN-γ and poly I:C simultaneously or IFN-γ and R848 simultaneously for 12–18 h. Subsequently, the cells were washed three times with PBS and detached from the wells using the 0.25% Trypsin-EDTA solution. The detached cells were collected by centrifugation at 100× *g* for 10 min and resuspended in the flow staining buffer. The pooled cells were stained by the PE anti-mouse CD86 antibody and DAPI. The stained cells were analyzed on the FACSAria™ Fusion Flow Cytometer.

### 2.7. Visualization of Dendritic Cell Activation

In addition to the quantitative flow cytometry analysis, dendritic cell activation was visualized by immunostaining and fluorescence imaging. In short, 2-well chamber slides were treated with 0.1% gelatin solution to facilitate cell adhesion. JAWSII dendritic cells (1 × 10^6^) were seeded into the wells and allowed to adhere overnight. The cells were placed in the serum starvation medium for 6 h and then treated with 100 µg/mL poly I:C and 10 ng/mL IFN-γ for 12–18 h. For optimum visualization, the cells were fixed to the 2-well chamber slide using 4% paraformaldehyde. To reduce nonspecific staining, the wells were blocked in 10% BSA in PBS. Subsequently, the cells were stained with 1 µg/mL PE anti-mouse CD86 antibody in 1% BSA and 1 µg/mL DAPI in PBS. Lastly, the slide was mounted and examined under an Axio Observer.Z1/7 microscope made by ZEISS (Jena, Oberkochen, Germany).

### 2.8. Exploring the Potential of IFN-γ, Poly I:C, and R848 Applied Simultaneously

To investigate the efficacy of all three molecules applied simultaneously in activating dendritic cells, another flow cytometry experiment was performed. Similarly, JAWSII dendritic cells (2 × 10^6^) were seeded in a 6-well plate and allowed to adhere overnight. The dendritic cells were placed in the serum starvation medium for 6 h and treated with IFN-γ, poly I:C, and R848 simultaneously for 12–18 h. Then, the cells were washed and detached from the wells using the 0.25% Trypsin-EDTA solution. The detached cells were collected by centrifugation at 100× *g* for 10 min and resuspended in the flow staining buffer. The pooled cells were stained by the PE anti-mouse CD86 antibody, and DAPI, and the stained cells were analyzed on the FACSAria™ Fusion Flow Cytometer. 

### 2.9. Statistical Analysis

Data are expressed as means ± standard error unless specified. Comparisons among multiple groups were conducted using one-way ANOVA followed by Tukey’s HSD test. Differences were considered significant when *p*-values were less than 0.05.

## 3. Results

### 3.1. TLR Expression by JAWSII Murine Dendritic Cells

JAWSII is an immortalized, immature, and myeloid-type dendritic cell line derived from the bone marrow of p53−/− C57BL/6 mice. In a previous study characterizing JAWSII dendritic cells, it was found that they share similar expression patterns of the key activation and/or maturation markers with bone marrow-derived dendritic cells (BMDCs). At resting state, JAWSII dendritic cells exhibited low expression of CD86 and CD40, moderate expression of MHC II, and high expression of CD11b, CD11c, CD80, MHC I, and ICAM-1/CD54. Upon stimulation by Chlamydia antigens, the expression levels of CD86, CD40, and MHC II all went up compared to the resting state [36]. The expression profile of these markers is very similar between JAWSII and BMDCs, rendering JAWSII an ideal model to study dendritic cell activation and maturation in addition to primary cells. More specifically, these expression patterns are more similar to those of type-2 conventional dendritic cells, which are often distinguished from the other types of dendritic cells by the expression of CD11b [37]. 

The expression of TLR3, 4, and 7 by JAWSII dendritic cells has been validated at the protein level in a previous study. Western blot analysis revealed that TLR3 and 7 were highly expressed in JAWSII cells at both resting and stimulated states. TLR4 was also expressed at a moderate level on the cell surface, as detected by fluorescence-activated cell sorting [38]. Although TLR8 was not mentioned in this study, it has been well-established that TLR8 is expressed in dendritic cells [39]. In this study, RT-PCR was performed to confirm the previously published knowledge using the primer pairs listed in Table 1. The primer pairs were selected according to the validation results provided by the PrimerBank database [40,41,42]. The amplicon sizes of the target genes of TLR3, 4, 7, and 8 are 162, 129, 207, and 109 base pairs, respectively. The RT-PCR results revealed that the mRNAs responsible for the expression of TLR3, 4, 7, and 8 were present in JAWSII dendritic cells, as depicted in Figure 1. In other words, the dendritic cells expressed the TLRs of interest; therefore, the cell line could be utilized to study dendritic cell activation by the TLR agonists. A complete gel image can be found in Appendix A.

### 3.2. Response Time of JAWSII Dendritic Cells to Stimulation

To determine the optimum time to treat JAWSII dendritic cells with the stimulators, the amount of cell activation was observed at various time points of stimulation. The dendritic cells were treated with 100 µg/mL poly I:C and 10 ng/mL IFN-γ simultaneously for 2, 4, 12, 18, or 24 h. The concentrations of poly I:C and IFN-γ were determined based on a previous study aimed to optimize the activation conditions for JAWSII dendritic cells [38]. One negative control group was included in the experiment, which was JAWSII dendritic cells treated with only the serum starvation medium for 24 h. The number of CD86-positive cells obtained by the negative control group was used to adjust the position of the gate but not shown in Figure 2. From Figure 2, the percentage of CD86-positive cells increased drastically from 2 to 12 h of stimulation and decreased from 18 to 24 h. Evidently, the amount of cell activation plateaued from 12 to 18 h, and it was most likely that the curve peaked at some point during this period. As a result, for the following flow cytometry experiments, the dendritic cells were treated with IFN-γ and/or the TLR agonists for 12–18 h to achieve the maximum cell activation level. 

### 3.3. Efficiency of IFN-γ or Individual TLR Agonist in Activating Dendritic Cells

To evaluate the efficiency of IFN-γ and the TLR agonists, poly I:C and R848, in activating dendritic cells, flow cytometry experiments were conducted. The concentrations of IFN-γ and poly I:C were determined based on a previous study in which 10 ng/mL IFN-γ and 100 µg/mL poly I:C were used to stimulate JAWSII dendritic cells [38]. In this study, concentrations centered around 10 ng/mL and near 100 µg/mL were selected for IFN-γ and poly I:C, respectively. A reference concentration of R848 in activating dendritic cells was not identified in the literature. Therefore, a relatively large span of concentrations of R848 ranging from 1 ng/mL to 10 µg/mL was investigated in this study. The percentage of CD86-positive cells in treatment groups relative to the control could be obtained through proper gating. Example results of flow cytometry depicting the gating strategy are shown in Figure 3A. The gates were placed to the right of the control cell population because PE fluorescence intensity would increase as more cells were activated by the stimulators. It is worth noting that other activation markers of dendritic cells, including CD80 and MHC II, were considered in the early stages of the study besides CD86. However, suitable antibodies were not identified to indicate the expression level of CD80 and MHC II accurately. A PE-linked anti-MHC I antibody was able to detect dendritic cell activation, but MHC I molecules are more ubiquitously expressed and not usually taken as an activation marker [43,44]. Therefore, CD86 was selected as the sole marker because it gives the most reliable and sensible indication of dendritic cell activation compared to the other markers. 

From the cytometric analysis, IFN-γ efficiently stimulated a considerable number of the dendritic cells. As illustrated in Figure 3B, 10 ng/mL IFN-γ achieved 33.4% cell activation, which was significantly higher than the negative control (*p* < 0.0001). In comparison, the TLR agonists by themselves could only activate a few compared to the control. The best TLR agonist treatment, 100 µg/mL poly I:C, could merely activate 6.0% of the dendritic cells, despite that it was statistically significantly different from the control (Figure 3C). In addition to the activation, the percentage of living cells was also analyzed, which could be indicative of possible toxic effects of the treatments. As shown in Figure 3D,E, IFN-γ was not likely harmful to the dendritic cells, as the percentage of living cells in its treatment groups was comparable to that of the control. However, R848 and poly I:C, at relatively high concentrations, might be slightly toxic to the dendritic cells. The percentages of viable cells of the groups treated with 100 ng/mL R848, 10 µg/mL R848, and 100 µg/mL poly I:C were 71.5%, 75.4%, and 81.6%, respectively. All were statistically significantly lower than that of the negative control group, which was 90.9% (*p* = 0.0003, 0.0036, and 0.0481, respectively).

### 3.4. Potency of IFN-γ and TLR Agonists as Dual, Complementary Systems

Another set of flow cytometry experiments was performed to explore whether the combination of IFN-γ (10 ng/mL) and a TLR agonist could further facilitate dendritic cell activation. In these experiments, the treatment of 10 ng/mL IFN-γ was taken as the control since it was the most potent, single molecule treatment obtained from the previous experiments. Interestingly, the amount of dendritic cell activation was elevated when the cells were stimulated by IFN-γ and a TLR agonist simultaneously. As shown in Figure 4A, 50 and 100 µg/mL poly I:C together with 10 ng/mL IFN-γ activated 52.7% and 59.1% of the dendritic cells, respectively. Both were significantly higher than the 33.4% CD86-positive cells yielded from 10 ng/mL IFN-γ (*p* = 0.0234 and 0.0004, respectively). Although statistical significance was not observed, most of the other combinatorial groups also achieved higher amounts of cell activation than the treatment of IFN-γ alone. The treatment of 100 µg/mL poly I:C with 10 ng/mL IFN-γ activated 59.1% of the dendritic cells, the highest among all the combinatorial groups. The difference between its group mean, and that of 10 ng/mL IFN-γ (25.7%) was more remarkable than the percentage of CD86-positive cells obtained by 100 ug/mL poly I:C (6.0%). These results suggested a potential synergistic effect between the two classes of molecules on dendritic cell activation, but further studies on the interaction between their stimulatory activities are needed.

Additionally, the combinatorial treatments did not seem toxic to the dendritic cells. As shown in Figure 4B, statistical significance was not observed when comparing the percentage of living cells of the treatment groups with that of the negative control. Even the groups stimulated by relatively high concentrations of R848 and poly I:C did not show any significant reduction in the number of living cells. Compared to the treatment groups shown in Figure 3B, IFN-γ seems to offset the toxicity of high concentrations of the TLR agonists. To visualize dendritic cell activation, immunostaining, and fluorescence imaging were carried out. As depicted in Figure 4C,D, cell nuclei were stained blue by DAPI, which preferentially stains dead cells [45], and CD86 proteins on the cell surface were stained red by the PE anti-mouse CD86 antibody. The negative control group did not show any red fluorescence except that of the background, indicating that CD86 was not expressed or the cells were not activated. In contrast, some cells stimulated by 100 µg/mL poly I:C and 10 ng/mL IFN-γ displayed bright red fluorescence mostly around their perimeters. A small quantity of red fluorescence might seem to dip into the cytoplasm of one or two cells. This could occasionally be due to background noise, overlapping of cells, or aggregation of the PE fluorescent dye. Another possible explanation is the incomplete transport or trafficking of newly synthesized CD86 proteins from the Golgi apparatus to the cell membrane [46]. Nonetheless, these images suggested that CD86 was expressed by the dendritic cells or that the cells were activated by the treatment of IFN-γ and poly I:C simultaneously. 

### 3.5. Dendritic Cell Reaction to IFN-γ, Poly I:C, and R848 Tri-Stimulator System

In addition to the IFN-γ-poly I:C and IFN-γ-R848 dual-stimulator systems, the efficiency of the IFN-γ-poly I:C-R848 tri-stimulator system in activating dendritic cells was also investigated. In this study, the amount of dendritic cell activation obtained by the three molecules simultaneously was compared to the dual-molecule treatments as well as the best-performing single stimulator treatment of 10 ng/mL IFN-γ. As depicted in Figure 5A, the results of flow cytometry revealed that the tri-stimulator treatment did not evoke a higher degree of dendritic cell activation than the dual-molecule treatments. It activated 32.3% of the dendritic cells, which was comparable to the 33.4% achieved by 10 ng/mL IFN-γ. The most prominent treatment group was still the dual-molecule system of 100 µg/mL poly I:C and 10 ng/mL IFN-γ. It activated 59.1% of the dendritic cells, which was significantly higher than the tri-stimulator treatment (*p* = 0.0011). From Figure 5B, the combinatorial treatment of relatively high concentrations of R848 and poly I:C led to a reduced percentage of living cells, which was 81.0%. It was statistically significantly lower than that of the negative control, which was 87.3% (*p* = 0.0222). In contrast, the treatment of the three molecules simultaneously obtained a percentage of living cells comparable to that of the negative control.

## 4. Discussion

It has been well documented that TLRs are expressed in innate immune cells, including dendritic cells, and some non-immune cells, such as fibroblasts [25]. Nevertheless, since JAWSII is an immortalized murine dendritic cell line [47,48], RT-PCR was performed to ensure that the cell line did not lose TLR expression at any point during its propagation. Among the four TLRs verified, TLR3 specifically detects double-stranded RNAs, which are usually found during viral replication [49]. Poly I:C is a synthetic double-stranded RNA, having one strand of all inosinic acid (I) and the other of all cytidylic acid (C). Upon recognition by TLR3, poly I:C triggers the TRIF-dependent signaling pathway, which essentially leads to a series of antiviral responses, such as the production of pro-inflammatory cytokines [50,51,52]. TLR7 and 8 are closely related in terms of their phylogeny and structure, and R848 is able to interact with both [53,54]. R848 is a synthetic imidazoquinoline compound, which upon binding to TLR7/8, stimulates the MyD88-dependent signaling pathway. The MyD88-dependent pathway leads to the activation of NF-κB, a transcription factor that induces the expression of a series of pro-inflammatory genes [55,56]. Despite the well-established immunostimulatory activities of the TLR agonists, surprisingly, poly I:C and R848 failed to activate dendritic cells to a significant extent. This could be, at least partially, attributed to the fact that TLR signaling pathways do not directly lead to APC activation. They do so indirectly by producing pro-inflammatory cytokines, such as type I interferons; type 1 interferons then activate APCs to enhance their antigen-presenting functions [25,57]. 

It might be a relatively lengthy process from TLR agonist stimulation to type I interferon-induced APC activation as a series of laborious signal transduction activities is involved. Therefore, the poor performance of the TLR agonists in activating dendritic cells could result from insufficient reaction time for the process to complete. In comparison, IFN-γ is much more potent than poly I:C and R848 in activating the dendritic cells. It has been demonstrated that IFN-γ is able to activate APCs, and activated APCs produce more IFN-γ [58]. As a result, activated APCs further stimulate neighboring cells, and this creates an autocrine loop by which APC activation is amplified and prolonged [59,60,61]. As a cytokine itself, IFN-γ directly acts on APCs and does not require lengthy signal transduction processes to take place. Therefore, a much higher amount of dendritic cell activation was observed in the IFN-γ treatment groups after a relatively short reaction time (12–18 h). As a reminder, this reaction time was determined based on the amount of CD86-positive cells when treated with 10 ng/mL IFN-γ and 100 µg/mL poly I:C simultaneously. The toxicity of the TLR agonists, R848 and poly I:C, has been previously reported [54,62,63,64]; however, IFN-γ might be able to offset it. Relatively high concentrations of R848 (100 ng/mL and 10 µg/mL) or poly I:C (100 µg/mL) together with 10 ng/mL IFN-γ did not escalate cell death compared to the control. This could potentially be explained by the fact that IFN-γ activated the dendritic cells in a timely manner, and the cells were immunologically active and defensive against R848 or poly I:C. 

The combination of either TLR agonists with IFN-γ resulted in elevated dendritic cell activation compared to what was achieved by them individually. This could potentially be interpreted as a complementary or synergistic effect between the two classes of molecules. It was found that in addition to pro-inflammatory cytokines, TLR agonists also promote the production of anti-inflammatory molecules such as IL-10. These molecules are involved in the negative feedback loops designed to regulate inflammation so that excessive or autoimmune reactions are prohibited [65,66]. In a previous study on macrophages, IFN-γ demonstrated promotive activity on TLR ligand-induced macrophage activation by suppressing the regulatory effect of IL-10 [67]. The pro-inflammatory cytokines generated from the TLR signaling pathways, such as type I interferons, would further act on and stimulate the APCs. Nonetheless, the combination of all three molecules, IFN-γ, poly I:C, and R848, did not yield a higher amount of dendritic cell activation. It is worth noting that prior to stimulation, the cells were placed in a serum starvation medium (99.8% MEM α and 0.2% FBS) for 6 h to synchronize them to the same cell cycle phase [35]. Considering the limitations in terms of nutrients and reaction time, the dendritic cells might have been overdriven by the stimulation of all three molecules simultaneously. Otherwise, assuming the dendritic cells could deal with all three molecules, more type I interferons might be created since different TLR signaling pathways were triggered by poly I:C and R848, respectively. However, high concentrations of type I interferons might antagonize the effect of IFN-γ on APC activation [57], thus leading to a fewer amount of dendritic cell activation compared to the two-molecule treatments. 

## 5. Conclusions

In conclusion, results presented in this study suggest that IFN-γ and TLR agonists could be applied as complementary systems to promote dendritic cell activation and the subsequent antigen presentation. IFN-γ not only activated a significant number of the dendritic cells by itself but also facilitated the TLR signaling pathways when administered concurrently with TLR agonists. To fully reveal a synergistic effect between the two classes of immunostimulants, further studies focusing on the mechanisms and interactions of their immunostimulatory activities need to be conducted. TLR agonists by themselves could merely activate a few of the dendritic cells speculatively because the TLR signal transduction processes take longer than the designated treatment time (12–18 h). Additionally, the IFN-γ-poly I:C-R848 tri-stimulator system did not enhance dendritic cell activation compared to the dual-molecule treatments. This could be the result of the cells being overdriven or type I interferons at certain high concentrations suppressing the immunostimulatory activity of IFN-γ. The best combinatorial treatment in terms of dendritic cell activation is 100 µg/mL poly I:C and 10 ng/mL IFN-γ. This formulation could be incorporated as immunoadjuvants into cancer immunotherapies or vaccines against infectious diseases or psychoactive compounds and further tested in animal models.

## Figures and Tables

**Figure 1 viruses-15-01198-f001:**
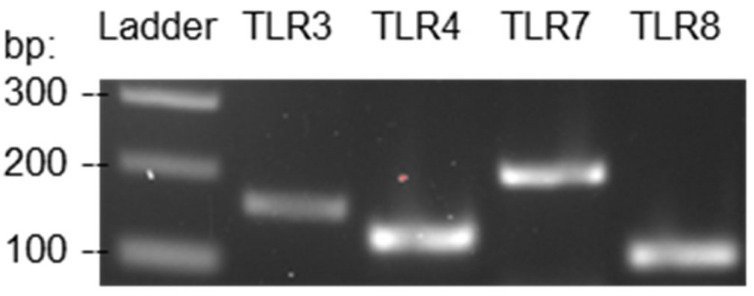
RT-PCR results authenticated the expression of TLR3, 4, 7, and 8 by JAWSII murine dendritic cells.

**Figure 2 viruses-15-01198-f002:**
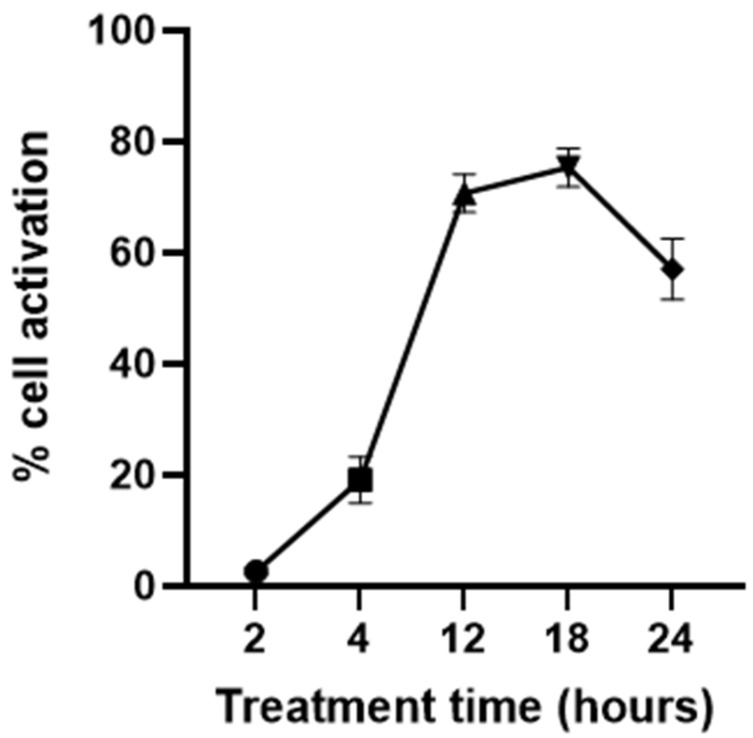
Dendritic cell response to the stimulation times of 2, 4, 12, 18, and 24 h (n = 3). Dendritic cells were stimulated by the treatment of 100 µg/mL poly I:C together with 10 ng/mL IFN-γ.

**Figure 3 viruses-15-01198-f003:**
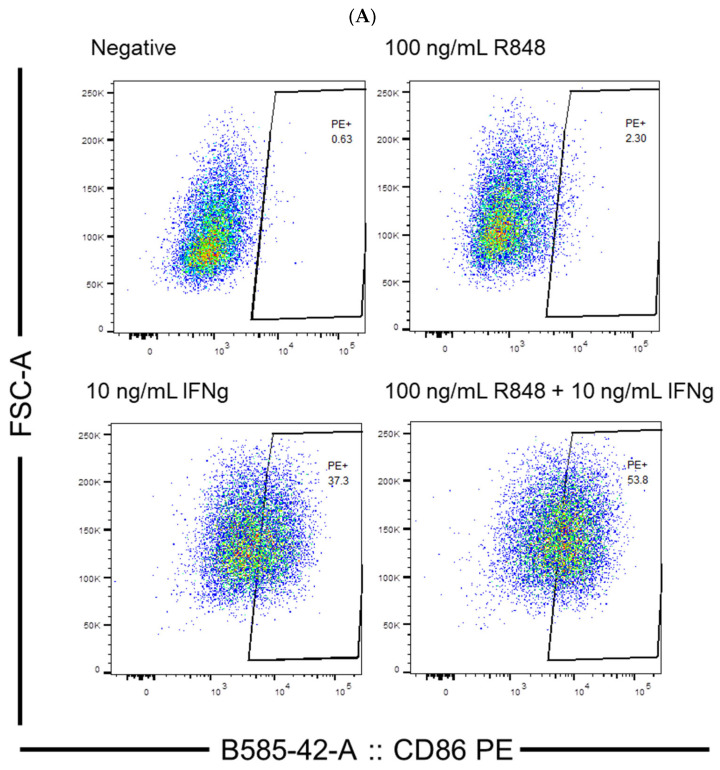
(**A**) Example results of flow cytometry depicting the gating strategy; (**B**) dendritic cell activation by IFN-γ (n = 13 for negative, n = 4 for 1 ng/mL IFN-γ, n = 9 for 10 ng/mL IFN-γ, and n = 4 for 100 ng/mL IFN-γ); (**C**) dendritic cell activation by TLR agonists (n = 10 for negative, n = 3 for R848 treatment groups, n = 4 for 20 and 50 µg/mL poly I:C, and n = 5 for 100 µg/mL poly I:C); (**D**) the percentage of living cells after treatment with IFN-γ, no significant difference was detected between any of the treatment groups and the negative control; (**E**) the percentage of living cells after treatment with TLR agonists. The negative control mentioned in Panel (**B**–**E**) was JAWSII dendritic cells treated with the serum starvation medium (99.8% MEM α and 0.2% FBS). * *p* ≤ 0.05, ** *p* ≤ 0.01, *** *p* ≤ 0.001, **** *p* ≤ 0.0001.

**Figure 4 viruses-15-01198-f004:**
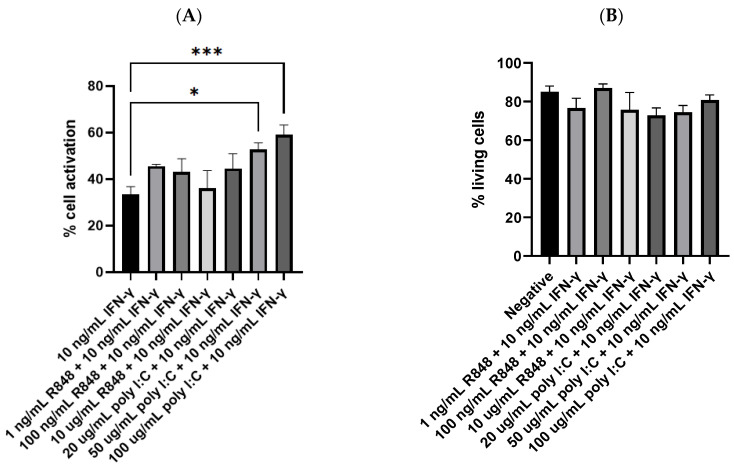
(**A**) Dendritic cell activation by IFN-γ or IFN-γ with TLR agonists (n = 9 for 10 ng/mL IFN-γ, n = 4 for all R848 together with IFN-γ groups, n = 4 for 20 and 50 µg/mL poly I:C together with IFN-γ groups, and n = 6 for 100 µg/mL poly I:C together with IFN-γ group); (**B**) the percentage of living cells after treatment with IFN-γ and TLR agonists simultaneously, no significant difference was detected between any of the treatment groups and the negative control (n = 10 for negative; n’s for the treatment groups are the same as in Figure 4A); (**C**) fluorescence image of negative control cells; (**D**) fluorescence image of cells treated with 100 µg/mL poly I:C and 10 ng/mL IFN-γ. The negative control mentioned in Panel (**B**) was JAWSII dendritic cells treated with the serum starvation medium (99.8% MEM α and 0.2% FBS). * *p* ≤ 0.05, *** *p* ≤ 0.001.

**Figure 5 viruses-15-01198-f005:**
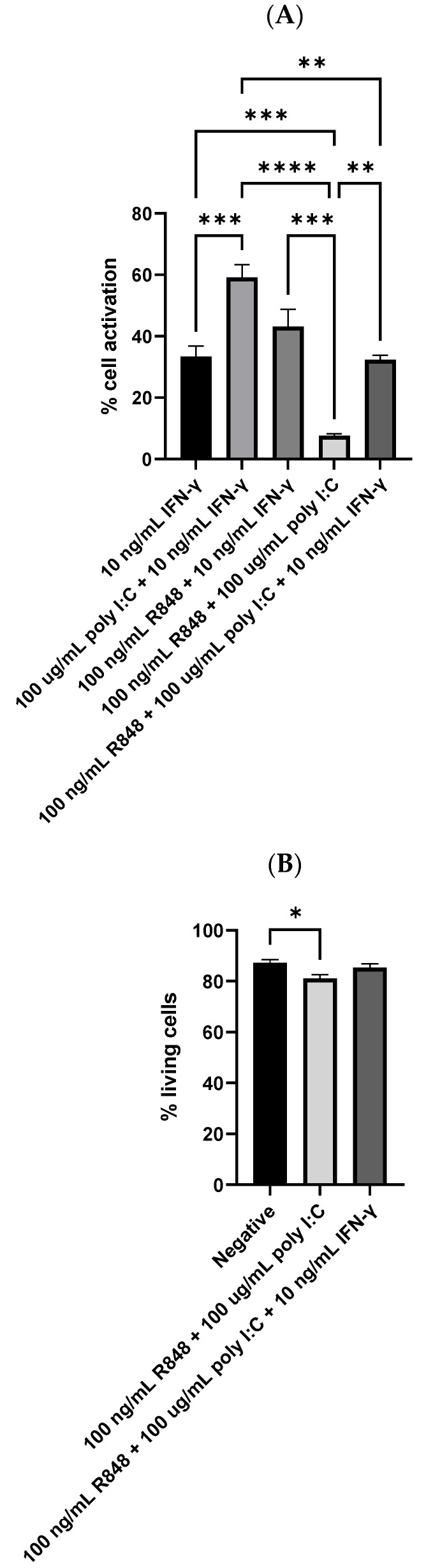
(**A**) Dendritic cell activation by IFN-γ, IFN-γ with one TLR agonist, or IFN-γ with both TLR agonists (n = 9 for 10 ng/mL IFN-γ, n = 6 for 100 µg/mL poly I:C in combination with 10 ng/mL IFN-γ, and n = 4 for the rest three groups) and (**B**) the percentage of living cells after treatment with both TLR agonists or IFN-γ with both TLR agonists (n = 4 for all groups). The negative control mentioned in Panel B was JAWSII dendritic cells treated with the serum starvation medium (99.8% MEM α and 0.2% FBS). * *p* ≤ 0.05, ** *p* ≤ 0.01, *** *p* ≤ 0.001, **** *p* ≤ 0.0001.

**Table 1 viruses-15-01198-t001:** List of primer pairs selected for RT-PCR on the target genes of TLR3, 4, 7, and 8.

Genes [*Mus musculus*]	NCBI Reference	Primer Sequences (5’-3’)		Size (bp)
		Forward	Reverse	
Tlr3	NM_126166.5	GTG AGA TAC AAC GTA GCT GAC TG	TCC TGC ATC CAA GAT AGC AAG T	162
	NM_001357316.1			
	NM_001357317.1			
Tlr4	NM_021297.3	ATG GCA TGG CTT ACA CCA CC	GAG GCC AAT TTT GTC TCC ACA	129
Tlr7	NM_001290755.1	ATG TGG ACA CGG AAG AGA CAA	GGT AAG GGT AAG ATT GGT GGT G	207
	NM_001290756.1			
	NM_133211.4			
	NM_001290757.1			
	NM_001290758.1			
Tlr8	NM_133212.3	GAA AAC ATG CCC CCT CAG TCA	CGT CAC AAG GAT AGC TTC TGG AA	109
	NM_001313760.1			
	NM_001313760.1			

## Data Availability

The data that support the findings of this study are available from the corresponding author upon reasonable request.

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
