# Peer review of "Efficiency of Interferon-γ in Activating Dendritic Cells and Its Potential Synergy with Toll-like Receptor Agonists"

_viruses, 2023, doi:10.3390/v15051198_

Round 1
Reviewer 1 Report
n this study, the researchers investigated the efficacy of interferon-γ, toll-like receptor agonists, and their combinations on dendritic cell activation in vitro. They discovered that INFγ may synergize with TLR agonists to activate dendritic cells. These findings could be of interest to researchers in the field of vaccine development and cancer immunotherapies. Some minor revisions are needed to further improve the manuscript:
- The title of the manuscript could be changed to "Potency of Interferon-γ in Activating Dendritic Cells In Vitro and Its Potential Synergy with Toll-like Receptor Agonists."
- The authors should briefly explain the concentrations of INFγ and TLR agonists used in this study.
- The spelling of IFNg and IFNγ should be made consistent throughout the manuscript.
- Statistical analyses are needed for Figure 3D and Figure 4B.
Author Response
We thank the insightful comments of the reviewer.
Comments: In this study, the researchers investigated the efficacy of interferon-γ, toll-like receptor agonists, and their combinations on dendritic cell activation in vitro. They discovered that INFγ may synergize with TLR agonists to activate dendritic cells. These findings could be of interest to researchers in the field of vaccine development and cancer immunotherapies. Some minor revisions are needed to further improve the manuscript:
Comments: The title of the manuscript could be changed to "Potency of Interferon-γ in Activating Dendritic Cells In Vitro and Its Potential Synergy with Toll-like Receptor Agonists."
Responses: We feel this current title better reflects the content of the manuscript, and thus the title of the manuscript was not changed.
Comments: The authors should briefly explain the concentrations of INFγ and TLR agonists used in this study.
Responses: The concentrations of IFN-γ and poly I:C used in this study were determined based on a previous study that aimed to optimize the activation conditions for JAWSII dendritic cells (Zapala et al., 2011). In the previous study, 10 ng/mL IFN-γ and 100 µg/mL poly I:C were applied to stimulate JAWSII dendritic cells. Therefore, concentrations centered around 10 ng/mL and near 100 µg/mL were selected for IFN-γ and poly I:C, respectively, in the current study. However, a reference concentration of R848 in activation dendritic cells was not found in the literature. As a result, a relatively large span of concentrations ranging from 1 ng/mL to 10 µg/mL was explored in the current study. Corresponding text addressing this issue has been added in the manuscript in lines 266-268 and 282-288.
Reference
Zapala, L., Drela, N., Bil, J., Nowis, D., Basak, G. W., & Lasek, W. (2011). Optimization of activation requirements of immature mouse dendritic JAWSII cells for in vivo application. Oncology reports, 25(3), 831–840.
Comments:The spelling of IFNg and IFNγ should be made consistent throughout the manuscript.
Responses: We thank the reviewer for this comment. IFN-γ is a generally accepted abbreviation for interferon-γ. It has been made consistent throughout the manuscript, specifically in Figures 3 to 5.
Comments:Statistical analyses are needed for Figure 3D and Figure 4B.
Responses:Statistical analyses were performed for the data presented in Figures 3D and 4B. They were not shown because no significant differences were detected between those treatment groups. Texts have been added to the figure legends to clarify the non-significance, as shown in lines 332-333 and 383-384.
Reviewer 2 Report
Good work on the introduction to get novice readers up to speed in limited space. The controls were well thought of and the experiments were sufficient to support the aim of the study and formulate conclusions.
Brief Summary:
With the stated aim(s) of the paper in mind, as I read through the paper it was apparent that the aims were indeed addressed. The article stated as its main aims to gather data demonstrating that dendritic cell activation and subsequent antigen presentation can be enhanced by the simultaneous treatment of interferon gamma (IFNg), thereby activating it’s related pathway via it’s cognate cell surface receptor, and with the treatment of poly I:C and resiquimod, both of which have toll like receptors (TLRs) intracellularly. Thus, the stimulation of dendritic cells via cell surface receptors and intracellular receptors were in place. Sound reasoning with examples demonstrating the promise IFNg holds as an adjuvant was provided, which I believe warrants further investigation, as shown in this study. Furthermore, the use of TLRs agonists to engage downstream signaling that produce type 1 interferons, coupled with the use of IFNg, is noted to potentially possess synergistic effects. Although no data was presented in regard to increased antigen presentation by dendritic cells, I find it a reasonable assumption (as the authors state) that the increased expression of CD86, which is a marker noted for dendritic cell activation, would correlate with increased antigen presentation. Potential toxicity effects to antigen presenting cells upon exposure of these powerful TLR and IFNg receptor agonists was presented as well, which is indeed an important consideration to hold.
General concept comments:
The manuscript was well written and supremely clear. It was well structured with sufficient background information to get novice readers up to speed on the biological and immunological concepts present in the study. The study generated compelling evidence to further consider cytokines as adjuvants to certain vaccines that don't quite elicit the immune response desired. I found the conclusions stated by the authors to be in-line with the findings of the study.
Author Response
We appreciate the support of this reviewer. Since no comments need to be addressed, no changes were made.